# Fingertip Whole Blood as an Indicator of Omega-3 Long-Chain Polyunsaturated Fatty Acid Changes during Dose-Response Supplementation in Women: Comparison with Plasma and Erythrocyte Fatty Acids

**DOI:** 10.3390/nu13051419

**Published:** 2021-04-23

**Authors:** Barbara J. Meyer, Cassandra Sparkes, Andrew J. Sinclair, Robert A. Gibson, Paul L. Else

**Affiliations:** 1School of Medicine, Molecular Horizons, University of Wollongong, Wollongong, NSW 2522, Australia; sparkesc@gmail.com (C.S.); pelse@uow.edu.au (P.L.E.); 2Illawarra Medical Research Institute, Wollongong, NSW 2522, Australia; 3Faculty of Health, Deakin University, Geelong, VIC 3220, Australia; Andrew.sinclair@deakin.edu.au; 4Department of Nutrition, Dietetics and Food, Monash University, Notting Hill, VIC 3168, Australia; 5Faculty of Sciences, School of Agriculture, Food and Wine, University of Adelaide, Adelaide, SA 5005, Australia; robert.gibson@adelaide.edu.au

**Keywords:** omega-3, erythrocyte, plasma, whole blood, dose response

## Abstract

The sensitivity of fingertip whole blood to reflect habitual dietary and dose-dependent supplemental omega-3 long-chain polyunsaturated fatty acid (n-3 LCPUFA) intake in premenopausal women was compared to that of venous erythrocytes and plasma fatty acids. Samples were obtained from women in a randomised, double-blind, placebo-controlled trial in which premenopausal women (*n* = 53) were supplemented with DHA-rich tuna oil capsules and/or placebo (Sunola oil) capsules (6 capsules per day) for 8 weeks to achieve doses of either 0, 0.35, 0.7 or 1.05 g/day n-3 LCPUFA. All blood biomarkers were very similar in their ability to reflect dietary n-3 LCPUFA intake (r = 0.38–0.46 for EPA and DHA intake), and in their dose-dependent increases in n-3 LCPUFA levels after supplementation (R^2^ = 0.41–0.51 for dose effect on biomarker EPA and DHA levels (mol %)). Fingertip whole blood is an effective alternative to erythrocytes and plasma as a biomarker n-3 LCPUFA intake in premenopausal women.

## 1. Introduction

Blood levels of omega-3 long-chain polyunsaturated fatty acids (n-3 LCPUFA), specifically eicosapentaenoic acid (EPA, 20:5n-3) and docosahexaenoic acid (DHA, 22:6n-3) are related to the risk of death from coronary heart disease (CHD) [1,2,3]. Therefore, measures of an individual’s n-3 LCPUFA status, such as the Omega-3 Index (EPA+DHA content in erythrocytes as a percentage of total fatty acids), show promise as a tool to reduce CHD risk [4]. More recently, n-3 LCPUFA levels have been shown to be associated with birth outcomes [5]. Moreover, it is well established that DHA plays an important role during pregnancy. In very early pregnancy, the mother mobilises DHA at the critical time of the neural tube closure (prior to 29 days of gestation) [6], as at that time there is a high demand for neural outgrowth that requires DHA [7]. The brain accrues DHA during gestation [8], and together with arachidonic acid (AA), DHA is required for proper neurological development. It has been shown that pregnant women with a low Omega-3 Index are at increased risk of preterm birth [9]. A meta-analysis of 70 trials including over 20,000 women showed that preterm birth and early preterm birth were reduced in those women receiving omega-3 supplementation compared to those women receiving placebo [5]. In Australia, the Department of Health’s Pregnancy Care Guidelines have been released (and endorsed by the National Health and Medical Research Council), and they state: “Omega-3 fatty acids Supplementation with omega-3 long-chain polyunsaturated fatty acids (800 mg docosahexaenoic acid [DHA] and 100 mg eicosapentaenoic acid [EPA] per day) may reduce the risk of preterm birth among women who are low in omega-3” [10].

Therefore, it has been postulated that the omega-3 status could also be used in pregnancy [11,12]. However, conventional fatty acid analysis is complex and time-consuming [13], therefore more simple and cost-effective methods are needed. The use of small volumes of blood from the fingertip is one such method [14,15,16,17,18,19,20]; however, the ability of fingertip whole blood to accurately reflect dietary n-3 LCPUFA intake is not well understood. Initial findings are that fingertip whole blood effectively reflects dietary fat intake [21], and its fatty acid profile correlates with factors such as gender, pregnancy, diet and smoking [14]. Fingertip whole blood levels of DHA can be used to differentiate between low- and high-fish consumers, and AA levels can be used to differentiate between low- and high-meat consumers [21]. While this was a significant and novel finding, the study was limited by the self-reporting of dietary intake. Moreover, the ability of fingertip whole blood to accurately reflect the fatty acid levels in blood fractions that are most often reported, such as plasma and erythrocytes, is lacking.

Few studies have investigated changes in fingertip whole blood n-3 LCPUFA status following controlled n-3 LCPUFA consumption [14,18]. Observation of washout in these studies showed that DHA has a long retention time in fingertip whole blood [14,18]. After high-dose (4.8 g/day) n-3 LCPUFA supplementation, n-3 LCPUFA were retained in fingertip whole blood longer than plasma, but not as long as erythrocytes [18]. However, comparisons between biomarkers from the study by Metherel et al. should be made with caution, as the method used for determining fingertip whole blood fatty acid composition is known to result in poor recovery of EPA and DHA [17]. Only one placebo-controlled trial has investigated the response of fingertip whole blood to longer-term n-3 LCPUFA supplementation [22], and this study was conducted in young children, not adults.

The objective of this study was to compare the ability of a fingertip blood to reflect dietary n-3 LCPUFA intake assessed using a validated food-frequency questionnaire (FFQ) [23], both in terms of its mole percentages (mol %) and concentrations (µg/mL blood) of fatty acids, to that of standard biomarkers such as plasma and erythrocytes in premenopausal women, controlling for menses. It also aimed to compare the sensitivity of each biomarker to dose-response n-3 LCPUFA supplementation in the range of normal dietary intake.

## 2. Materials and Methods

### 2.1. Study Design and Protocol

Blood samples were collected from women enrolled in a previously published double-blind, placebo-controlled, randomised controlled trial of parallel design [24]. Exclusion criteria for the study included women <18 or >40 years of age, and individuals with irregular menstrual cycles (<27 or >32 days), known existing cardiovascular disease (CVD), or those already consuming fish oil supplements who were unwilling to complete a 12-week washout period. Individuals with fasting plasma triglyceride levels <1.0 mmol/L were also excluded, as the study was designed to lower plasma triglyceride levels. Approval for the study protocol was granted by the Human Research Ethics Committee of the University of Wollongong (HE06/317), and written informed consent was obtained from subjects prior to commencement of the study. The clinical trial was registered in the Australian New Zealand Clinical Trials Registry (ID: ANZCTRN12607000566437).

Eligible subjects were randomly assigned to consume 6 capsules daily of oil, resulting in a dose of 0, 0.35, 0.7 or 1.05 g/day of n-3 LCPUFA (0 g EPA and 0 g DHA; 0.07 g EPA and 0.27 g DHA; 0.14 g EPA and 0.54 g DHA; 0.21 g EPA and 0.81 g DHA, respectively) for 2 menstrual cycles (approximately 8 weeks). Randomisation was controlled for age, BMI and contraceptive pill use, and there were no differences between systolic and diastolic blood pressures, total plasma cholesterol, dietary intake of energy, fat, protein and carbohydrate at baseline [24]. The n-3 LCPUFA was provided in the form of HiDHA tuna oil and/or Sunola oil placebo capsules (500 mg, supplied by Nu-Mega Ingredients, Brisbane, Australia). Each tuna oil capsule provided 135 mg DHA and 35 mg EPA, 107 mg monounsaturated fatty acids (MUFA), 135 mg saturated fatty acids (SFA), less than 10 International Units of vitamin A and 0.21% mixed natural tocopherols. Each Sunola capsule provided 355 mg MUFA, 55 mg SFA and 14 mg polyunsaturated fatty acids (PUFA), and contained no n-3 LCPUFA. The four different doses of n-3 LCPUFA were achieved by varying the proportion of active (HiDHA tuna oil) and placebo (Sunola oil) capsules. Subjects’ habitual intakes of PUFA were determined using a validated FFQ [23].

Subjects provided a fasting venipuncture and finger-prick blood sample after an overnight fast (>10 h) at baseline, and after approximately 8 weeks (2 menstrual cycles) of fish oil supplementation. Venous blood was collected into EDTA tubes and subjected to centrifugation at 4 °C for 10 min, followed immediately by separation of plasma and erythrocytes, which were then both stored at −80 °C for later fatty acid analysis. Fingertip whole blood samples were obtained by puncturing the fingertip using an automatic lancet (Medilance 2.0 mm, #450426); 100 µL of blood was collected using heparinised glass capillary tubes (Livingstone, Mascot, Australia) and immediately analysed for fatty acids. Clinic visits were conducted between days 3–5 of the menstrual/contraceptive pill cycle, as determined by counting from onset of menses.

### 2.2. Fatty Acid Analysis

Preparation of fatty acid methyl esters (FAMEs) from plasma was performed using the direct transesterification method of Lepage and Roy [25]. Briefly, 200 μL plasma was added to Teflon-lined 10 mL glass screw-cap tubes, followed by the addition of 2 mL of methanol/toluene (4:1 *v*/*v*) containing BHT as an antioxidant (0.01% *w*/*v*), and 200 μL of 0.2 mg/mL internal standard solution (heneicosanoic acid, 21:0; Sigma Aldrich, Castle Hill, Australia). Acetyl chloride (200 μL) was then added slowly using a positive displacement pipette, while vortexing tubes. Sealed tubes were then heated to 100 °C for 1 h and subsequently cooled, followed by addition of 5 mL of 6% potassium carbonate while vortexing. Tubes were then subjected to centrifugation at 2000× *g* for 10 min at 4 °C, and the supernatant (toluene phase) was collected and stored for gas chromatography analysis.

For erythrocyte fatty acids, aliquots of erythrocytes (400 μL) were thawed and re-suspended in a TRIS buffer (10 mM Bis Tris, 2 mM EDTA Na_2_, pH 7.2) in polycarbonate ultracentrifuge tubes (#3430, Nalgene, Rochester, NY, USA) at room temperature for 30 min. The samples were then spun in an ultracentrifuge at 315,000× *g* for 30 min at 4 °C (Beckman L-80 OPTIMA, Indianapolis, IN, USA) to pellet erythrocyte membranes. The supernatant was subsequently removed, and the erythrocyte membrane pellet was resuspended in 200 μL of distilled water. A fixed volume (150 μL) of the erythrocyte membrane suspension was used for direct transesterification of fatty acids using the method described for plasma fatty acids.

For fingertip whole blood, 100 μL of blood was pipetted into 10 mL glass screw-cap Teflon-lined tubes. To facilitate lysis of the blood cells, 100 μL of deionized water was added to the samples followed by vortexing, then allowing 30 min for the process to complete. During this time, a stock solution of transesterification reagents was prepared by combining 1.7 mL of methanol (containing 0.1 mg/mL BHT), 100 μL acetyl chloride and 100 μL internal standard (21:0 fatty acid, 0.1 mg/mL in toluene) per sample. After addition of 1.9 mL of stock solution to each sample, tubes were capped and heated to 100 °C for 1 h, and then cooled immediately. FAMEs were extracted by addition of 0.75 mL hexane, vortexed for 30 s and the upper organic (hexane) layer collected. This step was repeated once to optimise FAME extraction. Pooled hexane fractions were subsequently dried under nitrogen, reconstituted in 200 μL toluene and stored at −80 °C until analysed.

FAMEs were analysed by injecting μL of each sample in a gas chromatograph (GC 17A Shimadzu, Columbia, MD, USA) equipped with an autoinjector, 30 m FAME capillary column (0.25 mm internal diameter, Varian, Palo Alto, CA, USA) and flame ionization detector. Hydrogen was used as the carrier gas. Peaks were identified by comparison to known mixed standards (Nu-Chek Prep, Minnesota, MN, USA; Supelco F.A.M.E. mix C4-C24 (plus added DPA), #18919-1AMP, Sigma Aldrich, Castle Hill, Australia), and fatty acids were quantified using Shimadzu software (Class-VP 7.2.1 SP1, Kyoto, Japan). Fingertip whole blood, erythrocyte and plasma fatty acid mole percentages (mol %) and concentrations (µg/mL blood) were determined. Fatty acid concentrations were calculated using the peak area of an internal standard (heneicosanoic acid, 21:0). For erythrocytes and plasma, concentrations (expressed as µg/mL blood) were determined using an assumption of whole blood being composed of 45% erythrocytes and 55% plasma by volume.

The gas chromatography fatty acid output provides the area of the fatty acid peak detected. The percent area for each fatty acid is calculated as the area of the fatty acid divided by the total area of all the fatty acids detected and then multiplied by 100. The mol area is calculated as the % area divided by the molecular weight of the free/unesterified fatty acid. The mol % is calculated by dividing the mol area by the total mol area from all fatty acids and then multiplying by 100. The limit of quantification for a fatty acid is 0.05 mol %.

### 2.3. Statistical Analysis

JMP 5.1 statistical software (SAS, Cary, NC, USA) was used to perform all statistical analyses. The Shapiro–Wilk test was used to assess whether each variable fit a normal distribution. Non-normal variables were subsequently transformed using the log_10_ algorithm prior to statistical analyses. Unless stated otherwise, data presented in tables and figures are raw values.

Baseline differences between dose groups were examined using one-way ANOVA. Spearman’s correlations were used to assess relationships between dietary intakes and biomarker levels of PUFA. Post-supplementation comparisons were based on analysis of covariance (ANCOVA) standard least squares models with Tukey’s HSD analysis. In addition, a trend test for the effect of dose was performed for the 4 doses of n-3 LCPUFA using the dose as a continuous variable in a multiple regression model (standard least squares), baseline values as a covariate and the post-intervention values as the dependent variable. This multiple regression model was used to determine the statistical significance of dose (or baseline level) as a predictor of the post-intervention measure, as well as the combined predictive capacity of the two variables. Linear regression was used to determine the individual predictive capacity of each variable.

Of the 53 subjects included in analysis of plasma and erythrocyte fatty acids, PUFA questionnaire data was available for 45 subjects. Baseline and post-intervention fingertip whole blood samples were available for 49 subjects, 43 of whom completed the PUFA questionnaire. A value of *p* < 0.05 was considered statistically significant for all analyses.

## 3. Results

### 3.1. Relationships between n-3 LCPUFA Intake and Levels in Blood Biomarkers

The n-3 LCPUFA intakes from the habitual diet, in addition to DHA-rich tuna oil supplements (diet + supplemental), showed moderate to strong associations with their levels in fingertip whole blood, erythrocytes and plasma after supplementation (Table 1). For all biomarkers, the correlation coefficient was larger for DHA than EPA, except for erythrocytes when measured as µg/mL. Dietary + supplemental intake of EPA+DHA was strongly associated with post-supplemental EPA+DHA (mol %) to relatively the same degree in fingertip whole blood, plasma and erythrocytes (Figure 1). When expressed as mol %, the associations were slightly higher than when expressed as concentrations (ug/mL blood).

### 3.2. Response of Blood Biomarkers to Dose Response of n-3 LCPUFA Supplementation

The only individual PUFA to significantly increase in fingertip whole blood after 8 weeks of supplementation compared to baseline were EPA and DHA, as reported previously [26]. Hence, both total n-3 PUFA and n-3 LCPUFA (mol %) were significantly greater than the placebo group (for 0.7 and 1.05 g/day) and the 0.35 g/day group (for 1.05 g/day) (Table 2). While there was no significant increase in EPA levels after 0.7 g/day n-3 LCPUFA, after 1.05 g/day they were almost double that of the placebo and 0.35 g/day groups [26]. The increases in fingertip whole blood DHA levels were more substantial. After supplementation with 0.7 g/day and 1.05 g/day, DHA levels were approximately 45% and 66% higher than the placebo group, respectively [26]. Increases in fingertip whole blood n-3 LCPUFA levels (mol %) were countered by small but significant reductions in the levels of a number of SFA (14:0), MUFA (16:1n-7) and PUFA (20:3n-6). The largest reductions occurred in 16:1n-7, reflected by the significant reduction in total MUFA (Table 2).

The range of fingertip whole blood total n-3 was 3.1 to 9.2 mol %, and 23% of premenopausal women had fingertip whole blood total n-3 less than or equal to 4.1%.

### 3.3. Comparison of n-3 LCPUFA Dose Effect on Blood Biomarkers

The increases in fingertip whole blood, erythrocyte and plasma n-3 LCPUFA levels (mol %) following DHA-rich tuna oil supplementation were all dose-dependent (Table 2). The dose explained 39–41% of the variability in post-intervention fingertip whole blood and erythrocyte DHA levels (mol %), but it explained only 16% and 4% of its respective concentrations (ug/mL blood) (Table 3). The dose explained 36% and 51% of the variability in post-intervention plasma EPA and DHA (mol %), respectively, but it explained only 23% and 33% of its respective EPA and DHA concentrations (ug/mL blood) (Table 3). In all biomarkers, the effect of dose was slightly stronger for DHA than EPA, except for erythrocyte when measured as µg/mL. Dose was a predictor of total n-3 LCPUFA levels (mol %) to a similar extent as DHA in all biomarkers, despite there being no dose-dependent changes in n-3 DPA.

## 4. Discussion

This study provides the first comparative assessment of fingertip whole blood, erythrocytes and plasma fatty acids as biomarkers of habitual PUFA intake in premenopausal women and controlling for menses. The results of this study have shown that: (1) fingertip whole blood is comparable to erythrocytes and plasma in measuring dietary intakes of EPA and DHA; (2) expressing the fatty acid data as mol % shows higher correlations than expressing the data as concentrations (ug/mL blood); (3) all three biomarkers (fingertip whole blood, erythrocytes and plasma) responded in a dose-response manner to supplementation with tuna oil; and (4) the dose expressed as mol % explained the variability in post-intervention n-3 levels to a much greater extent than when expressed as concentrations (ug/mL blood). Therefore, fingertip whole blood is as robust an indicator of n-3 LCPUFA intake as venous plasma and erythrocytes, in terms of both habitual dietary intake and in response to supplementation with n-3 LCPUFA doses between 0 and 1.05g/day. These findings are particularly significant, given that they arose from a randomised, double-blind, placebo-controlled trial, in which habitual dietary intakes of PUFA also were related to their respective levels in a range of blood biomarkers within the same study population. Even though our study consumed higher n-3 LCPUFA than the national average, the dietary and supplemental n-3 LCPUFA intakes’ correlations with each biomarker (whole blood, erythrocytes and plasma) were linear across all intakes (i.e., lower and higher intakes), highly significant and explained approximately 50% of the variability for each biomarker.

A particular strength of the study is that the FFQ used was specifically geared towards measurement of dietary PUFA, is reproducible, and has been previously validated against blood biomarkers [23,27,28]. In fact, this PUFA FFQ has a very high degree of agreement with true dietary intake, with validity coefficients for EPA and DHA intake of 0.87–0.92 and 0.64–0.69 using plasma and erythrocytes as biomarkers, respectively [23]. The premenopausal women included in the current study consumed considerably more n-3 LCPUFA (median intake of 338 mg and interquartile range 186, 546) than the average dietary Australian woman, which is estimated to be 0.2 g/day [29]. Fish intake (including canned fish) contributed less than half of total n-3 LCPUFA intake. This finding highlights the importance of capturing n-3 LCPUFA intake data from foods other than fish and seafood, when assessing dietary n-3 LCPUFA intakes.

The strength of associations between dietary intake and whole blood levels of n-3 LCPUFA in the current study were similar to earlier reports. EPA and DHA intakes were previously found to be weakly-to-moderately associated with their levels in fasting venous whole blood (r = 0.22–0.23) and plasma (r = 0.28–0.31) [13]. Another study found that fingertip whole blood and dietary n-3 LCPUFA levels were not correlated in the full study cohort (*n* = 15); however, they were well correlated (r = 0.59) after removal of two subjects who consumed unusually high amounts of salmon within the data collection period [30]. These results utilised the collection of food records from a three-day period representative of the standard-diet residents of an aged care facility, and fatty acid analysis of food duplicates. A moderate association has also been reported between fingertip whole blood DHA levels and fish consumption [14]. Overall, the results presented here indicated that fasting fingertip whole blood was comparable to erythrocytes in its ability to reflect dietary EPA+DHA intake.

We observed no significant relationships between dietary intakes and biomarker levels of DPA. This is in agreement with the majority of previous studies that found plasma and erythrocytes to be good biomarkers for EPA and DHA, but not DPA [23,27,28,31,32]. This may be explained by the endogenous metabolism of n-3 LCPUFA, whereby membrane DPA levels are highly regulated and primarily serve as a substrate either for elongation to DHA or retroconversion to EPA [33,34].

The results of the present study also demonstrated that the sensitivity of fingertip whole blood to relatively small changes in n-3 LCPUFA intake was similar to that of erythrocytes and total plasma fatty acids. This was a particularly novel finding, given the paucity of data on how fingertip whole blood responds to increased n-3 LCPUFA intake in premenopausal women. Marangoni et al. [14] found that EPA and DHA levels in fingertip whole blood increased significantly after two weeks of supplementation with fish oil (0.65 g/day). This increase was comparable to that achieved with three weeks of salmon consumption, which provided considerably lower EPA+DHA intake. This was countered by a reduction in AA levels [14], unlike the present study wherein n-3 LCPUFA supplementation was mainly associated with reductions in MUFA and 20:3n-6 levels.

In terms of individual n-3 LCPUFA, dose was a stronger predictor of DHA than EPA levels. This was expected, given that the tuna oil supplement was much richer in DHA than EPA. While the dose effect on erythrocytes and fingertip whole blood was almost identical, it was slightly stronger in plasma. This is consistent with the greater magnitude of change in EPA and DHA observed in plasma fatty acids. The strong similarity in biomarker responses to supplementation was particularly significant, given that the doses provided in this study were much lower than those used in many previous n-3 LCPUFA supplementation trials. The lower end of the spectrum of doses is where one may expect to see a more differential response of the biomarkers, as saturation levels may not be reached in all biomarkers. In fact, the dose levels provided here are easily achievable by increasing intake of oily fish and seafood, functional foods enriched with n-3 LCPUFA [35] or commonly recommended doses of commercially available fish oil supplements. A positive finding was that the dose effect in fingertip whole blood was a good reflector of the combined responses of erythrocytes and plasma fatty acids.

This study also highlights the importance of n-3 levels in premenopausal women, especially if they become pregnant. Pregnant women that have fingertip whole blood n-3 levels less than or equal to 4.1% early in their pregnancy are at increased risk of early preterm birth [9]. In the current study, 23% of premenopausal women had a fingertip whole blood level less than or equal to 4.1%. This demonstrates the need to screen pregnant women in early gestation for n-3 levels.

The limitation of this study was the small sample size; however, despite this, significant results were seen, despite the highly variability in response to each dose [26]. This study was conducted in premenopausal women and hence is generalizable for this population, but further research is warranted for men and postmenopausal women.

Overall, the results of the present study provided strong evidence that fingertip whole blood is a valid biomarker of n-3 LCPUFA intake in premenopausal women. Not only did it display a similar relationship with habitual and supplemental intake to plasma and erythrocytes obtained by venipuncture, the dose-response incorporation of n-3 LCPUFA in fingertip whole blood was also similar to these biomarkers. This demonstration of equivalent sensitivity to low dose n-3 LCPUFA supplementation means that fingertip whole blood samples could be used to monitor progressive improvements in blood levels in a clinical setting. Given that there are numerous methodological and cost-saving advantages to analysis of n-3 LCPUFA status in fingertip whole blood, such as reduced time, reduced sample volume and reagents/solvents required, and higher throughput, this approach has the capacity to increase the likelihood of widespread application of n-3 LCPUFA testing in clinical practice, while ensuring that the strength of the biomarker is not compromised.

## 5. Conclusions

Fingertip whole blood is a valid biomarker of n-3 LCPUFA habitual intake and supplemental intake in premenopausal women. This cost-effective method of fingertip whole blood assessment should be used to screen pregnant women during early gestation to identify those at risk of preterm birth.

## Figures and Tables

**Figure 1 nutrients-13-01419-f001:**
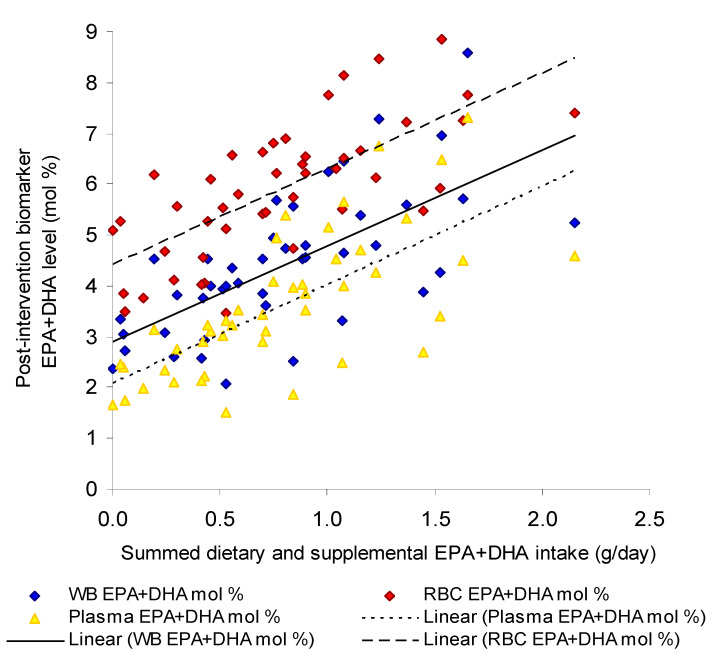
Scatterplot of individual summed dietary and supplemental intakes of EPA+DHA (g/day) and corresponding fingertip whole blood (R^2^ = 0.52 *p* < 0.0001, *n* = 43), plasma (R^2^ = 0.56 *p* < 0.0001, *n* = 45) and erythrocyte (R^2^ = 0.56 *p* < 0.0001, *n* = 45) EPA+DHA levels at post-intervention (mol %).

**Table 1 nutrients-13-01419-t001:** Spearman’s correlation coefficients (r_s_) for associations between the sum of dietary (as measured by FFQ) and supplemental intakes of n-3 LCPUFA and post-intervention blood biomarker levels (mol %) and concentrations (µg/mL blood) (*n* = 45 for erythrocytes and plasma, *n* = 43 for fingertip whole blood).

Dietary Plus Supplemental Fatty Acid Intakes	Fingertip Whole Blood		Erythrocytes		Plasma	
	Mol %	Concentration (µg/mL Blood)	Mol %	Concentration (µg/mL Blood)	Mol %	Concentration (µg/mL Blood)
20:5n-3	0.61 **	0.56 **	0.67 **	0.38 *	0.60 **	0.54 **
22:6n-3	0.69 **	0.62 **	0.72 **	0.34 *	0.76 **	0.63 **
EPA + DHA	0.72 **	0.63 **	0.75 **	0.34 *	0.75 **	0.65 **
Total n-3 LCPUFA	0.65 **	0.58 **	0.65 **	NS	0.71 **	0.61 **

* *p* < 0.05, ** *p* < 0.0001. Total n-3 LCPUFA is EPA, n-3 DPA plus DHA. NS: nonsignificant.

**Table 2 nutrients-13-01419-t002:** Changes in fingertip whole blood fatty acid levels (mol %) after supplementation with 0–1.05 g/day n-3 LCPUFA from DHA-rich tuna oil.

	0 g/day (*n* = 11)		0.35 g/day (*n* = 15)		0.7 g/day (*n* = 11)		1.05 g/day (*n* = 12)	
	0 Weeks	8 Weeks	0 Weeks	8 Weeks	0 Weeks	8 Weeks	0 Weeks	8 Weeks
14:0	1.3 (1.2, 1.5)	1.3 (1.17, 1.5)	1.2 (1.0, 1.5)	1.1 (0.98, 1.3)	1.2 (0.93, 1.4)	0.98 (0.84, 1.1) *	1.3 (1.1, 1.5)	0.96 (0.83, 1.1) *
16:0	26 (25, 26)	26 (25, 26)	25 (25, 26)	25 (24, 26)	25 (24, 26)	25 (24, 26)	25 (25, 26)	25 (24, 26)
18:0	9.8 (9.3, 10)	9.7 (9.1, 10)	9.9 (9.5, 10)	10.0 (9.8, 10)	10.0 (9.6, 10)	10.8 (9.9, 12) *	9.9 (9.4, 10)	10.4 (9.9, 11)
22:0	0.74 (0.62, 0.89)	0.72 (0.60, 0.87)	0.78 (0.66, 0.93)	0.77 (0.65, 0.91)	0.69 (0.53, 0.89)	0.75 (0.60, 0.95)	0.73 (0.61, 0.88)	0.73 (0.59, 0.90)
24:0	1.3 (1.1, 1.5)	1.2 (1.0, 1.5)	1.4 (1.2, 1.6)	1.3 (1.1, 1.6)	1.3 (0.96, 1.6)	1.2 (0.99, 1.6)	1.3 (1.1, 1.5)	1.3 (1.1, 1.6)
16:1n-7	2.3 (1.9, 2.9)	2.4 (2.0, 3.0)	2.1 (1.7, 2.6)	1.9 (1.6, 2.3)	2.3 (1.7, 2.6)	1.6 (1.2, 2.2) *	2.4 (2.1, 2.7)	1.7 (1.5, 2.0) *
17:1n-7	0.81 (0.66, 0.99)	0.88 (0.69, 1.14)	0.90 (0.79, 1.0)	1.04 (0.94, 1.2)	0.75 (0.61, 0.91)	0.99 (0.87, 1.1)	0.83 (0.72, 0.95)	0.82 (0.59, 1.2)
18:1n-9	17 (16, 18)	18 (17, 19)	18 (17, 18)	17 (16, 18)	18 (17, 19)	17 (16, 18)	17 (16, 18)	17 (15, 18)
24:1n-9	1.5 (1.3, 1.8)	1.5 (1.3, 1.7)	1.6 (1.4, 1.8)	1.6 (1.4, 1.8)	1.4 (1.1, 1.8)	1.4 (1.1, 1.8)	1.5 (1.3, 1.8)	1.6 (1.3, 1.9)
18:2n-6	21 (19, 22)	20 (19, 22)	21 (19, 22)	21 (20, 23)	21 (20, 23)	21 (19, 24)	21 (20, 22)	21 (20, 22)
20:3n-6	1.7 (1.4, 2.0)	1.8 (1.6, 1.9)	1.6 (1.5, 1.8)	1.5 (1.4, 1.7)	1.7 (1.5, 1.9)	1.3 (1.1, 1.5) *^,#^	1.6 (1.4, 1.8)	1.3 (1.1, 1.4) *^,#^
20:4n-6	9.4 (8.7, 10)	8.7 (8.3, 9.2)	9.0 (8.4, 9.6)	8.9 (8.2, 9.6)	8.5 (8.0, 9.0)	8.2 (7.8, 8.7)	8.9 (8.5, 9.3)	8.8 (8.4, 9.2)
22:4n-6	1.0 (0.90, 1.1)	0.92 (0.79, 1.1)	0.92 (0.83, 1.0)	0.86 (0.76, 0.97)	0.86 (0.75, 0.99)	0.79 (0.72, 0.87)	0.82 (0.73, 0.92)	0.75 (0.67, 0.84)
22:5n-3	0.91 (0.82, 1.0)	0.91 (0.80, 1.0)	0.95 (0.86, 1.0)	0.90 (0.82, 0.99)	0.96 (0.83, 1.10)	0.88 (0.80, 0.96)	0.96 (0.83, 1.1)	0.91 (0.83, 1.0)
SFA	39 (39, 40)	39 (39, 40)	39 (39, 40)	39 (38, 40)	39 (38, 40)	40 (38, 41)	39 (38, 40)	39 (38, 40)
MUFA	22 (21, 23)	23 (22, 24)	23 (22, 24)	22 (21, 23)	23 (22, 24)	21 (20, 22) *	22 (21, 23)	21 (20, 22) *
PUFA	39 (38, 40)	38 (36, 39)	38 (36, 39)	39 (37, 40)	38 (36, 40)	39 (37, 41)	38 (37, 40)	40 (39, 41)
n-6	34 (32, 35)	33 (31, 34)	33 (32, 34)	34 (32, 35)	33 (32, 35)	33 (31, 35)	33 (32, 34)	33 (31, 34)
n-3	4.8 (4.3, 5.3)	4.7 (4.1, 5.2)	4.5 (4.1, 4.9)	5.1 (4.7, 5.6)	4.7 (4.2, 5.1)	6.1 (5.4, 6.8) *	4.9 (4.2, 5.8)	6.9 (6.0, 8.0) *^,#^
n-3 LCPUFA	4.3 (3.9, 4.8)	4.3 (3.8, 4.8)	4.1 (3.7, 4.5)	4.7 (4.3, 5.1)	4.2 (3.8, 4.7)	5.6 (5.0, 6.4) *	4.5 (3.8, 5.3)	6.5 (5.6, 7.6) *^,#^

Values are mean (approximate 95% CI) calculated on the log scale and then back-transformed to raw scale for presentation (*n* = 49). Differences between dose groups were compared using log-transformed data with ANCOVA followed by post hoc Tukey’s HSD tests. * Significantly different from placebo (*p* < 0.05); ^#^ significantly different from 0.35 g/day (*p* < 0.05). Fatty acids constituting < 0.5 mol % of total fatty acids not shown. For EPA and DHA values, refer to [26]. Abbreviations: SFA: saturated fatty acids; MUFA: monounsaturated fatty acids; PUFA: polyunsaturated fatty acids; n-3 LCPUFA: total n-3 long-chain PUFA is EPA, n-3 DPA plus DHA.

**Table 3 nutrients-13-01419-t003:** Coefficient of determination (R^2^) for fingertip whole blood, erythrocyte and total plasma fatty acids where dose was a significant predictor of post-intervention fatty acid levels (expressed as mol % or µg/mL blood; *n* = 49 for fingertip whole blood, *n* = 53 for plasma and erythrocytes).

Dose Dependent	Fingertip Whole Blood		Erythrocytes		Plasma	
	R^2^ (mol %)	R^2^ (µg/mL)	R^2^ (mol %)	R^2^ (µg/mL)	R^2^ (mol %)	R^2^ (µg/mL)
14:0	0.21 **	0.09 *	0.09 **	0.05 *	0.16 *	0.17 *
18:0	0.09 *	NS	NS	NS	NS	NS
16:1n-7	0.10 **	0.06 *	NS	NS	0.13 **	0.14 *
18:1n-9	0.10 **	NS	0.08 **	NS	0.13 *	0.12 *
24:1n-9	NS	NS	NS	NS	0.12 **	NS
20:3n-6	0.27 ***	0.14 **	0.12 **	NS	0.10 *	0.12 *
20:4n-6	NS	NS	0.18 **	NS	NS	NS
20:5n-3	0.31 **	0.15 *	0.29 **	0.08 *	0.36 ***	0.23 **
22:6n-3	0.41 ***	0.16 **	0.39 ***	0.04 **	0.51 ***	0.33 ***
Total n-3 LCPUFA	0.39 ***	0.13 **	0.42 ***	0.02 *	0.51 ***	0.31 **

*** *p* < 0.0001, ** *p* < 0.01, * *p* < 0.05. Total n-3 LCPUFA is EPA, n-3 DPA plus DHA. NS: nonsignificant.

## Data Availability

The data sets generated and/or analyzed during the current study are not publicly available, but are available from the corresponding author upon reasonable request.

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
