# Peer review of "Fingertip Whole Blood as an Indicator of Omega-3 Long-Chain Polyunsaturated Fatty Acid Changes during Dose-Response Supplementation in Women: Comparison with Plasma and Erythrocyte Fatty Acids"

_nutrients, 2021, doi:10.3390/nu13051419_

Round 1

Reviewer 1 Report

I have reviewed the manuscript by Meyer and colleagues, entitled "Fingertip whole blood as an indicator of omega-3 long chain polyunsaturated fatty acid changes during dose response supplementation in women: Comparison with plasma and erythrocyte fatty acids."

This is an important contribution to the ongoing and evolving debate how to best measure the status of EPA and DHA in individuals, permitting making well-founded conclusions about the relationship between omega-3 intake and physiological outcomes. This study is important in that respect by taking a methodological approach to identifying the most practical and scientifically sound measurement approach to doing that. Overall the manuscript is well presented. A number of comments is provided below which the authors can address to further improve their manuscript and provide some additional clarity to the future readers.

Comments and suggestions:

Line 92 – Please also indicate the exact amount of EPA and of DHA provided with each dose (in addition to already stated dose of n-3 LCPUFA). See below comments regarding the need to define which omega-3 fatty acids are included in the n-3 LCPUFA classification.

Line 94 – As for the sunola oil, please provide a brief description of the composition of the main fatty acid classes of the HiDHA tuna oil used in this study. The HiDHA oil contains 135 mg DHA and 35 mg DHA per 500 mg capsule, but what is the composition of the remaining 330 mg? The sunola oil is mentioned to contain primarily MUFA, and apparently the authors consider this of importance – How much MUFA is present in the HiDHA oil?

Line 95 – Please provide information on the antioxidant type(s) and concentration used to stabilize both oils. Information on the type of material used for the capsule is also welcome (not usually reported but of interest) – e.g. type of gelatin, vegetable polymer, or other.

Line 150 – C21:0 is heneicosanoic acid, not heneicosaenoic acid.

Line 152 – It would be good if the authors could introduce a short paragraph here describing exactly how mol% was calculated. Since expression in mol% is central to this study, it is important to provide an exact description of mol% expression, and assist authors citing your work express in mol% in the same manner. For example - is mol% calculated as percentage of total fatty acids? Are molecular weight calculations of the fatty acids based on the free/unesterified fatty acid form or on the fatty acid-methyl esters actually detected? What is the limit of quantification for fatty acids that are included and summed in the calculation of mol%?

Line 178: “ For all biomarkers, this association was stronger for DHA than EPA”. This comment can only be made if a statistical test was used to compare correlations. These tests exist, the simplest of which are based on the Fisher r-to-z transformation.

Line 196 - While there was no significant increase in EPA levels after 0.7g/day n-3 LCPUFA, after 1g/day they were almost double that of the placebo and 0.35g/day groups”.  Each tuna oil capsule contains 35 mg EPA and 135 mg DHA. Assuming that most of the n3-LCPUFA is EPA+DHA, that means that 0.7 g/day n-3 LCPUFA means about 144 mg/day EPA and 555 mg/day of DHA. According to (https://www.ncbi.nlm.nih.gov/pmc/articles/PMC4808841/), the median n-3 LCPUFA intake (for Australians) is 119 mg/day, so the 0.7 g/day treatment provides more EPA than the average diet. The authors need to address why such high level of supplementation is not detected by the whole blood method. Is the reason biological or analytical? It would be useful to at least say something about the lower limit of detection for this method. If the reason is biological, this is fascinating. If analytical, then the failure to detect 144 mg/day of EPA is a limitation of the method.

Minor corrections for the presence of other n-3 LCPUFA species probably apply.

Line 221 - In all biomarkers the effect of dose was slightly stronger for DHA than EPA”.  A test to back this up is needed. Or, at least, a more detailed explanation of how this was figured - the R2 for DHA-erythrocytes-ug/mL is smaller than for EPA. Please rewrite.

Line 233 – “…whole blood is comparable to erythrocytes and plasma in measuring dietary intakes of EPA and DHA; “. Some further discussion is welcome here, since these intakes are higher than usual dietary intakes.

Line 251 – “The pre-menopausal women included in the current study consumed considerably more n-3 LCPUFA than the average dietary Australian woman, which is estimated to be 0.2g/day. The article needs to disclose the average intake in this cohort.

Table 2 - For clarity, it would be good if you can define the difference between n-3 PUFA and n-3 LCPUFA, for example in the Method section. Which n-3 species are not considered “long-chain” by the authors (for example alpha-linolenic acid and stearidonic acid)?

Secondly, most of the table is presented with 2 significant digits, but some of the confidence intervals have 3. Consistency would be preferred. An additional significant figure would be good to indicate for 18:1n9, as the current rounding makes it impossible to see why the 8 week 1.05 g/day result is different from placebo.

Thirdly - Multiple hypothesis correction would be useful - there are so many tests, and the p-value cutoff is 0.05, so one expects a number of false positives. It would be good to either adjust the p-value cutoff using a Bonferroni correction or similar, or at least to use different symbols for p <0.05, p<0.01.

Typos:

Line 182 – associations

Author Response

Reviewer 1

I have reviewed the manuscript by Meyer and colleagues, entitled "Fingertip whole blood as an indicator of omega-3 long chain polyunsaturated fatty acid changes during dose response supplementation in women: Comparison with plasma and erythrocyte fatty acids."

This is an important contribution to the ongoing and evolving debate how to best measure the status of EPA and DHA in individuals, permitting making well-founded conclusions about the relationship between omega-3 intake and physiological outcomes. This study is important in that respect by taking a methodological approach to identifying the most practical and scientifically sound measurement approach to doing that. Overall the manuscript is well presented. A number of comments is provided below which the authors can address to further improve their manuscript and provide some additional clarity to the future readers.

Comments and suggestions:

Line 92 – Please also indicate the exact amount of EPA and of DHA provided with each dose (in addition to already stated dose of n-3 LCPUFA). See below comments regarding the need to define which omega-3 fatty acids are included in the n-3 LCPUFA classification.

A: The values have been added (see new lines 93 and 94).

Line 94 – As for the sunola oil, please provide a brief description of the composition of the main fatty acid classes of the HiDHA tuna oil used in this study. The HiDHA oil contains 135 mg DHA and 35 mg DHA per 500 mg capsule, but what is the composition of the remaining 330 mg? The sunola oil is mentioned to contain primarily MUFA, and apparently the authors consider this of importance – How much MUFA is present in the HiDHA oil?

A: We have deleted the emphasis on sunola oil being primarily MUFA and added the additional details to HiDHA as requested. The sentence has been changed to:

“n-3 LCPUFA was provided in the form of HiDHA™ tuna oil and/or Sunola oil placebo capsules (500 mg, supplied by Nu-Mega Ingredients, Australia). Each tuna oil capsule provided 135mg DHA and 35mg EPA, 107mg monounsaturated fatty acids (MUFA), 135mg saturated fatty acids (SFA), less than 10 International Units of vitamin A and 0.21% mixed natural tocopherols. Each Sunola capsule provided 355mg MUFA, 55mg SFA and 14mg polyunsaturated fatty acids (PUFA), and contained no n-3 LCPUFA (see new lines 100-104).

Line 95 – Please provide information on the antioxidant type(s) and concentration used to stabilize both oils. Information on the type of material used for the capsule is also welcome (not usually reported but of interest) – e.g. type of gelatin, vegetable polymer, or other.

A: As stated in above, less than 10 International Units of vitamin A and 0.21% mixed natural tocopherols (see new lines 100-104).

Line 150 – C21:0 is heneicosanoic acid, not heneicosaenoic acid.

A: this has been corrected (see new lines 124 and 158).

Line 152 – It would be good if the authors could introduce a short paragraph here describing exactly how mol% was calculated. Since expression in mol% is central to this study, it is important to provide an exact description of mol% expression, and assist authors citing your work express in mol% in the same manner. For example - is mol% calculated as percentage of total fatty acids? Are molecular weight calculations of the fatty acids based on the free/unesterified fatty acid form or on the fatty acid-methyl esters actually detected? What is the limit of quantification for fatty acids that are included and summed in the calculation of mol%?

A: A small paragraph describing exactly how mol% was calculated has been added to the manuscript as suggested (see new lines 161-167).

Yes the mol% calculated as percentage of the total fatty acids. The molecular weight calculations of the fatty acids based on the free/unesterified fatty acid form. During the fatty acid process, all fatty acids become methylated, including the internal standard. The limit of quantification is 0.05 mol%.

Line 178: “ For all biomarkers, this association was stronger for DHA than EPA”. This comment can only be made if a statistical test was used to compare correlations. These tests exist, the simplest of which are based on the Fisher r-to-z transformation.

A: Thank you for pointing this out. We have changed the sentence to the following: “For all biomarkers, the correlation coefficient was larger for DHA than EPA except for erythrocytes when measured as µg/mL.” (see new lines 194-195).

Line 196 - While there was no significant increase in EPA levels after 0.7g/day n-3 LCPUFA, after 1g/day they were almost double that of the placebo and 0.35g/day groups”.  Each tuna oil capsule contains 35 mg EPA and 135 mg DHA. Assuming that most of the n3-LCPUFA is EPA+DHA, that means that 0.7 g/day n-3 LCPUFA means about 144 mg/day EPA and 555 mg/day of DHA. According to (https://www.ncbi.nlm.nih.gov/pmc/articles/PMC4808841/), the median n-3 LCPUFA intake (for Australians) is 119 mg/day, so the 0.7 g/day treatment provides more EPA than the average diet. The authors need to address why such high level of supplementation is not detected by the whole blood method. Is the reason biological or analytical? It would be useful to at least say something about the lower limit of detection for this method. If the reason is biological, this is fascinating. If analytical, then the failure to detect 144 mg/day of EPA is a limitation of the method.

A: Thank you for this excellent point that you have raised and I will explain that the reason is biological.

In our previously publication (Sparkes et al 2020) we compared the changes of erythrocytes, plasma and whole blood EPA, DHA, and EPA plus DHA in response to the various doses. Significant differences were seen in plasma EPA at 0.7g and 1g doses, but both erythrocyte and whole blood, EPA was not significantly different at 0.7g but was significantly different at the 1g dose. We know that changes in plasma occurs rapidly and prior to changes in erythrocytes, as plasma reflects recent dietary intakes, whilst erythrocytes reflect changes over months. Erythrocytes live for 120 days and this study intervention was for 8 weeks (i.e. 56 days), hence the lack of significance at 0.7g. However, 1g dose was high enough to achieve significant EPA changes in both erythrocytes and whole blood after 56 days. Given that whole blood is approximately 45% erythrocytes and 55% plasma, one would expect that whole blood would not be the same as plasma and this is what we see.

We have not added any of this biological explanation to the current manuscript as it is referring to the data that was published in 2020 (Sparkes C, Sinclair AJ, Gibson RA, Else PL, Meyer BJ. High variability in erythrocyte, plasma and whole blood EPA and DHA levels in response to supplementation Nutrients 2020, 12, 1017; doi:10.3390/nu12041017).

We have however, added the limit of quantification for fatty acids (see new lines166-167), as the other reviewer wanted more information regarding the calculation of mol% as well as quantification. 

Minor corrections for the presence of other n-3 LCPUFA species probably apply.

A: Thank you for your comment.

Line 221 - In all biomarkers the effect of dose was slightly stronger for DHA than EPA”.  A test to back this up is needed. Or, at least, a more detailed explanation of how this was figured - the R2 for DHA-erythrocytes-ug/mL is smaller than for EPA. Please rewrite.

A: Thank you for pointing this out. We have re-written the sentence by adding “except for erythrocyte when measured as µg/mL” to the sentence: “In all biomarkers the effect of dose was slightly stronger for DHA than EPA, except for erythrocyte when measured as µg/mL.” (see new lines 194-195).

Line 233 – “…whole blood is comparable to erythrocytes and plasma in measuring dietary intakes of EPA and DHA; “. Some further discussion is welcome here, since these intakes are higher than usual dietary intakes.

  1. Thank you for the suggestion of further discussion regarding the comparability of whole blood to erythrocytes and plasma in measuring dietary intakes of EPA and DHA. We have added the R2 and P values to figure 1, each biomarker explains approximately 50% of the variability, and all are highly significant. We considered adding further discussions to this point, but given the correlations are linear across all intakes (i.e. lower and higher intakes), we just added the following comment: “Even though our study consumed higher n-3 LCPUFA than the national average, the dietary and supplemental n-3 LCPUFA intakes’ correlations with each biomarker (whole blood, erythrocytes and plasma) were linear across all intakes (i.e. lower and higher intakes), highly significant and explained approximately 50% of the variability for each biomarker.” (See new lines 265-269)

Line 251 – “The pre-menopausal women included in the current study consumed considerably more n-3 LCPUFA than the average dietary Australian woman, which is estimated to be 0.2g/day. The article needs to disclose the average intake in this cohort.

A: The median intakes and interquartile range of n-3 LCPUFA have been added (see new line 276).

Table 2 - For clarity, it would be good if you can define the difference between n-3 PUFA and n-3 LCPUFA, for example in the Method section. Which n-3 species are not considered “long-chain” by the authors (for example alpha-linolenic acid and stearidonic acid)?

A: For clarity the “Total n-3 LCPUFA” has been defined as ‘Total n-3 LCPUFA is EPA, n-3 DPA plus DHA’ and this definition has been added to Tables 1, 2 and 3.

Secondly, most of the table is presented with 2 significant digits, but some of the confidence intervals have 3. Consistency would be preferred. An additional significant figure would be good to indicate for 18:1n9, as the current rounding makes it impossible to see why the 8 week 1.05 g/day result is different from placebo.

A: Thank you for pointing out the significant digits in Table 2, which we have corrected to 2 significant digits. We incorrectly had 18:1n-9 highlighted as being significantly different but we have corrected this in Table 2.

Thirdly - Multiple hypothesis correction would be useful - there are so many tests, and the p-value cutoff is 0.05, so one expects a number of false positives. It would be good to either adjust the p-value cutoff using a Bonferroni correction or similar, or at least to use different symbols for p <0.05, p<0.01.

A: All tables 1 to 3 and figure 1 have the P values listed with different symbols for p <0.05, p<0.01 and p<0.0001

Typos:

Line 182 – associations

A: Thank you for picking up this spelling mistake which has been corrected (see new line 198).

Reviewer 2 Report

Introduction:  

  • The introduction states the clinical importance of omega-3 fatty acid status in premenopausal women.  
  • The authors demonstrate that fingertip whole blood analysis would be a useful tool because it is simple, requires small volumes of blood, and cost-effective. 
  • Authors review current literature available on fingertip whole blood analysis, highlight previous research limitations, and identify the gaps 
  • The research purpose of the paper was clearly stated.  

Research Design: 

  • Authors address type of study and eligibility criteria, along with institutional approval and consent process 
  • It is not clear if this is a registered clinical trial. Also, it is not clear if this data is being used from a clinical trial they conducted and presenting the data differently or using data from a different clinical trial.  

Methods: 

  • Address how quickly blood samples were processed for storage after the initial blood draw. 
  • Line 115, there ia missing number after the decimal point. 
  • Extraction methods for plasma, erythrocytes, and fingertip whole blood are sync and detailed. 

Results: 

  • There is not a demographics table included. 
  • Section 3.2 It becomes unclear of the groups that are being compared around line 194 to 197. 
  • The sections are clearly delineated for discussing each result section. 

Discussion/Conclusion:   

  • The discussion is well organized and appropriately presents the results. 
  • The discussion is missing a limitations section to address issues of sample size.  
  • It is not clear how generalizable these findings are outside of the study population. 

Author Response

Reviewer 2

Introduction:  

  • The introduction states the clinical importance of omega-3 fatty acid status in premenopausal women.  
  • The authors demonstrate that fingertip whole blood analysis would be a useful tool because it is simple, requires small volumes of blood, and cost-effective. 
  • Authors review current literature available on fingertip whole blood analysis, highlight previous research limitations, and identify the gaps.  
  • The research purpose of the paper was clearly stated.  

A: thank you for these comments.

Research Design: 

  • Authors address type of study and eligibility criteria, along with institutional approval and consent process 
  • It is not clear if this is a registered clinical trial. Also, it is not clear if this data is being used from a clinical trial they conducted and presenting the data differently or using data from a different clinical trial.  

A: This study is from a registered clinical trial. The clinical trial was registered in the Australian and New Zealand Clinical Trial Registration ID: ANZCTRN12607000566437 (see new lines 90-91).

For clarity, we have added the words “previously published” in the sentence “Blood samples were collected from women enrolled in a previously published double-blind, placebo-controlled, randomized controlled trial of parallel design [24].” (see new lines 81-82).

Methods: 

  • Address how quickly blood samples were processed for storage after the initial blood draw. 

A: the samples were processed immediately and we have clarified this – see new lines 112 and 116

  • Line 115, there is a missing number after the decimal point. 

A: Thank you very much for pointing this out and it has been corrected. (see new line 124)

  • Extraction methods for plasma, erythrocytes, and fingertip whole blood are sync and detailed. 

A:  Thank you for your comment

Results: 

  • There is not a demographics table included. 

A: As this study is part of a previously published study (which we have now clarified), we cannot re-publish the demographics table. However, we have added the following “and there were no differences between systolic and diastolic blood pressures, total plasma cholesterol, dietary intake of energy, fat, protein and carbohydrate at baseline [24]”  to this sentence “Randomisation was controlled for age, BMI and contraceptive pill use and there were no differences between systolic and diastolic blood pressures, total plasma cholesterol, dietary intake of energy, fat, protein and carbohydrate at baseline [24].” (see new lines 96-98)

  • Section 3.2 It becomes unclear of the groups that are being compared around line 194 to 197. 

A: To clarify this, we have added the words “compared to baseline” to the sentence: “The only individual PUFA to significantly increase in fingertip whole blood after 8 weeks of supplementation compared to baseline were EPA and DHA as reported previously [26].” (see new line 211)

The sentence “Hence, both total n-3 PUFA and n-3 LCPUFA (mol %) were significantly greater than the placebo group (for 0.7 and 1.05g/day) and the 0.35g/day group (for 1.05 g/day) (Table 2).” Does state that the comparison is to the placebo group, so we have not changed this sentence. The legend underneath Table 2 shows the comparisons using different symbols, i.e. *Significantly different from placebo (p<0.05), #Significantly different from 0.35 g/day (p<0.05).

  • The sections are clearly delineated for discussing each result section. 

 A: Thank you for this comment.

Discussion/Conclusion:   

  • The discussion is well organized and appropriately presents the results. 

A: Thank you for these comments.

  • The discussion is missing a limitations section to address issues of sample size.  

A: Limitations has been added (see new lines 330-331)

  • It is not clear how generalizable these findings are outside of the study population. 

A: A comment about generalizable has been added to the limitations section (see new lines 331-333)